# A Review on the Structure-Response-Efficacy Optimization of Ultrasound-Responsive Micro/Nanobubbles for Cancer Therapy

**DOI:** 10.3390/pharmaceutics17111378

**Published:** 2025-10-24

**Authors:** Yuting Yang, Yuan Cheng, Zhiguang Chen, Yanjun Liu

**Affiliations:** Department of Ultrasound, The First Hospital of China Medical University, Shenyang 110001, China; yangyuting202205@163.com (Y.Y.); 18704249615@163.com (Y.C.)

**Keywords:** ultrasound micro/nanobubbles, targeted delivery, Structure-Response-Efficacy framework, acoustic optimization, theranostic integration

## Abstract

Ultrasound-responsive micro/nanobubbles (MNBs) are promising tools for targeted cancer therapy due to their controllable acoustic activation and real-time imaging. Despite extensive research, the quantitative relationship between bubble structure, acoustic response, and therapeutic efficacy remains poorly understood. This knowledge gap hinders parametric design and clinical standardization. This review summarizes recent advances from an engineering perspective, highlighting how structural parameters—such as size, shell, gas core, and ligand density—affect acoustic sensitivity and drug release. Furthermore, the roles of microfluidic electroporation and cell membrane coating are discussed in terms of controllable fabrication and preservation of biological functions, highlighting their significance for reproducible and predictable therapies. In conclusion, this review establishes a “Structure-Response-Efficacy (S-R-E)” framework to summarize the core relationships between structural design and acoustic modulation. We propose an engineering strategy based on a standardized parameter system to guide the predictable design and clinical translation of ultrasound-based theranostic platforms.

## 1. Introduction

Cancer remains a leading cause of mortality worldwide. While chemotherapy is a mainstay of treatment, its effectiveness is limited by poor tumor selectivity and severe systemic toxicity [1]. Nanodelivery systems were developed to address this challenge by leveraging the enhanced permeability and retention (EPR) effect for passive targeting [2]. However, these systems face formidable physical and biological barriers. On one hand, The dense stromal matrix and high interstitial fluid pressure in the tumor microenvironment (TME) often result in a drug delivery efficiency below 1% [3]. On the other hand, even when drugs reach tumor cells, multidrug resistance (MDR) mechanisms, such as P-glycoprotein (P-gp) efflux pumps, rapidly expel them, diminishing local therapeutic efficacy [4].

Among various strategies using external physical stimuli, ultrasound combined with micro/nanobubbles (MNBs) stands out for its minimal invasiveness and precise external controllability [5]. Ultrasound-induced cavitation can transiently permeabilize physical barriers in tumors, while MNBs, acting as acoustic-responsive carriers, facilitate on-demand, high-intensity drug release at the ultrasound focus. Notably, in vitro studies using drug-resistant cancer cell models have demonstrated that this localized, “burst-like” drug delivery, mediated by sonoporation, effectively overcomes MDR and restores the cytotoxic effects of chemotherapeutic agents [6].

While earlier reviews have covered fabrication, acoustic mechanisms [7], and clinical translation [8], they often lack a systematic analysis of the key engineering parameters that determine therapeutic outcomes.

To address this gap, we adopt an engineering-centric perspective and propose a “Structure-Response-Efficacy” (S-R-E) analytical framework. This framework explores how MNB structural design—including material selection, gas core type, and bioinspired functionalization [9]—influences acoustic response and its connection to in vivo pharmacokinetics and therapeutic efficacy. This review aims to provide theoretical guidance for the rational design and performance optimization of efficient and safe acoustic-responsive drug delivery systems, thereby advancing their clinical translation.

## 2. Engineering the Structural Design of Micro/Nanobubbles

The structural parameters of MNBS underpin the S-R-E framework because they determine both acoustic response and drug delivery efficacy. Structural optimization aims to balance stability and acoustic responsiveness under different conditions. Figure 1 summarizes the key engineering aspects, including core components, design trade-offs, and fabrication methods. This section systematically elaborates on the core structural components, surface functionalization, and advanced fabrication techniques.

### 2.1. Core Structural Elements: Size, Shell, and Gas Core

The bubble size, gas core composition, and shell material collectively influence the stability, cavitation mode, and energy transfer efficiency of MNBs in vivo. Due to their size difference, MBs (1–10 μm) and NBs (<1 μm) show distinct oscillation patterns. MBs can generate stable cavitation at low-frequency ultrasound (0.5–2 MHz), effectively enhancing local membrane permeability. NBs penetrate deeper tissues, making them suitable for targeting deep tumors. However, they require higher acoustic thresholds and stricter ultrasound parameters [10]. Recent research has further elucidated the quantitative relationship between bubble size and sonoporation. Under 1 MHz ultrasound at pressures of 250–500 kPa, bubbles with a radius of approximately 4.7 μm produced the highest membrane permeabilization efficiency—5 to 30 times greater than that of 2.1 μm bubbles. This size-dependent effect diminished as the pressure exceeded 500 kPa [11], demonstrating that bubble size can be tuned for precise control over the acoustic response intensity.

The gas core composition directly affects the dissolution rate and acoustic stability of the bubble. Commonly used gases include sulfur hexafluoride (SF_6_), perfluorobutane (C_4_F_10_), and nitrogen (N_2_). Among these, perfluorocarbon gases significantly extend circulation time and enhance stability due to their low solubility and slow diffusion rates [12]. The choice of gas not only regulates bubble lifespan but also influences acoustic backscatter intensity and imaging duration.

The shell material determines the bubble’s mechanical properties and acoustic sensitivity. Lipid monolayer shells exhibit excellent acoustic responsiveness and can be activated at low acoustic pressures; however, their stability is poor, making them prone to collapse or aggregation in the bloodstream. Polymer shells (e.g., PLGA) offer high mechanical strength, prolonging circulation time and improving in vivo integrity, but they require higher acoustic pressures to induce cavitation [13,14]. Protein-based shells (e.g., albumin) provide a balance between biocompatibility and acoustic sensitivity [12]. To further harmonize these properties, Dwivedi et al. designed a “liposome-microbubble conjugate,” which increased intratumoral drug concentration by 3.4-fold compared to the control group in animal models and significantly enhanced tumor suppression [15].

Additionally, bubble stability can be further optimized by incorporating cholesterol or through PEGylation. Cholesterol enhances membrane fluidity and shear resistance, while PEGylation effectively reduces immune recognition, prolongs circulation time, and improves passive accumulation [10]. These material-based strategies provide structural and functional support for deep-tissue drug delivery while ensuring biocompatibility.

### 2.2. Surface Functionalization Engineering: Targeting and Immune Evasion

Surface modification is a critical engineering step for modulating the in vivo stability, recognition, and delivery efficacy of MNBs [16]. By incorporating functional molecules onto the shell, a dual optimization of immune evasion and targeted recognition can be achieved [17].

Within the S-R-E framework, the first layer of control from surface modification comes from the type and density of ligands. Ligands recognize tumor-associated receptors (e.g., vascular endothelial growth factor receptor 2 (VEGFR2)”, “arginine–glycine–aspartic acid (RGD)”, “human epidermal growth factor receptor 2 (HER2)”, “epidermal growth factor receptor (EGFR)), enhancing bubble retention in tumor blood vessels [10]. However, Harpster et al. found that in RGD-modified microbubbles, an increase in surface ligand density led to a concurrent rise in particle size and surface tension. This, in turn, decreased their acoustic sensitivity and elevated the cavitation threshold, thereby weakening targeting efficiency [18]. Thus, surface modification impacts both binding affinity and acoustic behavior by changing shell mechanics. The linkage chemistry also influences the spatial conformation and exposure of ligands. Navarro-Becerra et al. noted that using PEG linkers or extended spacer arms can improve the accessibility of binding sites [10].

To prolong circulation lifetime and reduce immune clearance, PEGylation is the most common strategy. Attaching PEG chains to the bubble surface shields surface antigens and reduces clearance by the mononuclear phagocyte system, extending circulation time from 5 min to 30 min [19]. However, Zhang et al. discovered that excessively long or dense PEG chains can mask targeting ligand recognition sites, leading to reduced binding efficiency with VEGFR2 antibodies [20]. Therefore, PEGylation must be carefully balanced to enhance immune stealth without overly compromising targeting.

Overall, surface modification strategies require a delicate balance among stability, acoustic sensitivity, and targeted recognition. These advanced design concepts are exemplified in Figure 2, which demonstrates both the precise fabrication and magnetic control of microbubbles via microfluidics (Figure 2a–c) and a ‘smart’ pH-responsive system designed to activate within the tumor microenvironment (Figure 2d).

### 2.3. Advanced Design Strategies: Biomimetic Membrane Coating and Microfluidic Fabrication

Conventional microbubble fabrication methods often suffer from wide size distributions and significant batch-to-batch variability, failing to meet the demands of precision medicine. Consequently, “biomimetic microbubbles” and microfluidic technologies have gained prominence.

Biomimetic microbubbles utilize natural cell membranes (e.g., from red blood cells or tumor cells) as shell materials, endowing them with biological properties such as immune evasion, self-recognition, and long circulation. Hu et al. pioneered this approach by coating polymer nanoparticles with red blood cell membranes, creating a new class of biomimetic delivery platforms [9]. This “cell membrane coating” technology directly integrates the complex structures of native membrane proteins onto the nanoparticle surface, conferring superior physiological compatibility [22].

Microfluidic technology has shown significant advantages in the engineering of biomimetic systems. Rao et al. developed a low-stress biomimetic fabrication strategy based on microfluidic electroporation. By applying short electrical pulses to transiently increase membrane permeability, they efficiently loaded magnetic nanocores into red blood cell membrane vesicles [23]. This method surpasses traditional techniques in terms of membrane protein preservation, particle uniformity, and scalability. Recent studies highlight that microfluidic electroporation offers optimal performance in membrane integrity, batch-to-batch consistency, and material utilization. It allows precise control of membrane thickness, core–shell ratio, and size, forming the basis for a standardized fabrication–structure–function system [24].

In summary, the progression from the concept of membrane biomimicry to its engineering realization via microfluidic electroporation represents a complete technological development chain. The introduction of microfluidics not only enables the controllable fabrication of biomimetic microbubbles but also provides a foundation for standardizing the S-R-E framework [24].

### 2.4. Comparative Analysis of Microbubbles and Nanobubbles: A Clinical Application Perspective

In the field of ultrasound-responsive carriers, MBs and NBs are the two most fundamental tools. Although both use a gas core for acoustic response, their size difference greatly affects their in vivo behavior, acoustic properties, and clinical applications. Therefore, a systematic comparison is crucial for rationally designing the “structure” based on specific therapeutic needs. This section compares and contrasts MBs and NBs across several key dimensions and discusses their respective advantages and challenges in clinical translation [25] (see Table 1).

## 3. Engineering the Acoustic Response Behavior

A well-designed structure gives MNBs specific physical traits that determine their acoustic behavior under ultrasound. These responses link structural design to biological efficacy. This section outlines key physical mechanisms and how they can be controlled by engineering parameters.

### 3.1. Core Physical Mechanisms: Cavitation and Sonoporation

Ultrasound-mediated cavitation is the key physical basis for MNB-assisted drug delivery. When exposed to an acoustic field, bubbles undergo periodic expansion and contraction in response to alternating pressure. This behavior manifests in two primary modes: stable cavitation and inertial cavitation [26,27]. At lower acoustic pressures (typically 0.3–0.8 MPa), bubbles undergo stable, controllable oscillations. When oscillating near a cell membrane, the resulting push-pull effects and shear stress can create transient, reversible pores of 100–300 nm, a phenomenon known as sonoporation [26]. This pressure range is often termed the “sonoporation window,” as it significantly enhances cellular drug uptake while maintaining cell viability [27].

When the acoustic pressure exceeds a certain threshold (e.g., >0.8 MPa), bubbles undergo violent, uncontrollable expansion followed by instantaneous collapse, which is known as inertial cavitation [26,27,28]. This process releases high-energy shockwaves and microjets. Although these forces can also create pores in cell membranes, they often lead to irreversible cell damage or death [5,28,29]. Therefore, precisely controlling the acoustic response to remain within the “sonoporation window” is a prerequisite for achieving efficient and safe drug delivery.

### 3.2. Engineering Modulation for Spatiotemporal Control of Release

Two main factors control the acoustic response of MNBs: structural properties and external acoustic field parameters. Precise modulation of these parameters enables spatiotemporally controlled drug release.

Acoustic field parameters are the most direct means of engineering the response mode. Research has systematically identified optimal ranges for frequency, pressure, and duty cycle. For example, Chuang et al. reported optimal drug release and cell-killing effects at 1 MHz, 0.6 MPa, and a 50% duty cycle [13]. This shows that adjusting external parameters can keep the response within the optimal stable cavitation range.

The precise control of acoustic response is aimed at achieving on-demand drug release. Stable cavitation typically induces a sustained, slow leakage of the drug from the bubble shell. In contrast, inertial cavitation causes the complete rupture of the bubble, triggering a burst release of the payload [28]. This ultrasound-triggered release, occurring at a specified time and location (the acoustic focus), can significantly increase local drug concentrations in the tumor. For instance, research by De Cock et al. confirmed that ultrasound-activated bubbles could increase doxorubicin retention in the tumor region by 3- to 5-fold [28]. When combined with the inherent contrast imaging capability of some bubbles, a closed-loop therapeutic process of “imaging-guided, on-demand activation, and efficacy feedback” can be realized. This represents a crucial link connecting acoustic response to final therapeutic efficacy.

## 4. In Vivo Biological Efficacy and Clinical Applications

Controllable acoustic responses trigger a series of biological effects. This section explains how these mechanisms contribute to therapeutic efficacy, validating the S-R-E framework in practice. These effects begin with overcoming microscopic biological barriers, which in turn optimizes the macroscopic in vivo behavior of drugs and has ultimately been validated in early clinical settings.

### 4.1. Core Efficacy: Overcoming Biological Barriers and Optimizing Pharmacokinetics

Ultrasound-microbubble technology directly helps overcome biological barriers in drug delivery. At the cellular level, the sonoporation effect described in Section 3 provides an efficient transmembrane pathway for drugs, particularly for large macromolecular nucleic acids. Recent studies on ultrasound-responsive cavitation nuclei [30] and research by Helfield et al. confirmed the reversibility and safety of these nanoscale pores [31], while Escoffre et al. demonstrated that this method can effectively increase the intracellular concentration of the chemotherapy drug doxorubicin [32]. This changes the drug delivery paradigm from a reliance on slow endocytosis to a physically mediated transmembrane transport.

At the tissue level, cavitation acts on the physical barriers of the TME. By generating microstreaming and shear forces, it can transiently loosen the extracellular matrix and improve local blood perfusion [33,34,35], a finding validated in a 3D tumor model constructed by Zhao et al. [36]. Furthermore, this physical intervention can modulate the physiological state of the TME. For example, a study by Han et al. found that Ultrasound-Targeted Microbubble Destruction (UTMD) could ameliorate tumor hypoxia and acidosis [37]. This not only enhances drug penetration but also improves the tumor’s physiological response to therapy.

The successful breach of local barriers leads to an optimization of systemic drug pharmacokinetics. Structural designs, such as the ‘microbubble-liposome’ system by Dwivedi et al. [15], help prevent premature drug release during circulation. This enables ultrasound-triggered site-specific delivery, reducing off-target toxicity. However, the study by Chuang et al. also highlighted that the compatibility between the physicochemical properties of the drug (e.g., hydrophilicity/hydrophobicity) and the carrier structure is a key variable determining final release efficiency [13]. This highlights the critical role of structural design in determining therapeutic efficacy within the S-R-E framework.

### 4.2. Advanced Applications and Clinical Translation

Beyond drug delivery, this technology also supports theranostic applications. MNBs can serve as ultrasound contrast agents to enable real-time monitoring of drug release during treatment [38]. The recent introduction of artificial intelligence (AI) offers new possibilities for adaptive parameter control and personalized medicine [39]. Two recent clinical studies mark the transition of ultrasound-microbubble technology from preclinical research to clinical protocols that offer tangible benefits to patients.

In the field of radiotherapy, a phase I clinical trial by Moore-Palhares et al. demonstrated that MR-guided focused ultrasound combined with microbubbles could safely achieve an objective response rate as high as 83% in breast cancer patients [40], revealing its potential as a radiotherapy “sensitizer.”

In chemotherapy, a phase II clinical trial conducted by Han et al. in advanced pancreatic cancer showed that adding diagnostic-grade ultrasound and microbubbles to a standard chemotherapy regimen extended the median overall survival of patients from 6.1 months to 9.1 months [41]. This result indicates that the technology can be effective in clinical practice even without complex focused ultrasound equipment, showcasing its broad application potential.

These studies confirm that through the systematic engineering of structure, response, and efficacy, the theoretical advantages observed in the laboratory can be translated into clinical benefits for cancer patients.

## 5. Challenges and Future Outlook

### 5.1. Current Challenges

#### 5.1.1. Structural Design Challenge: The Inherent Conflict Between Stability and Acoustic Sensitivity

Structural design involves trade-offs between key performance parameters. For example, increasing shell flexibility can enhance acoustic sensitivity but reduces stability in the bloodstream [42]. Research by Boissenot et al. showed that reinforcing the shell with polymers (like PLGA) improves stability but necessitates higher acoustic pressures to trigger drug release [29]. Similarly, increasing the density of surface targeting ligands can improve recognition but may also decrease the acoustic response by increasing membrane tension [18]; meanwhile, PEG chains used to prolong circulation may mask these targeting ligands [20]. Currently, no universal design principle exists to balance these conflicting requirements.

#### 5.1.2. Data Heterogeneity and Lack of Standardization

A key challenge in this field is the significant variability in data across studies, along with the lack of standardized protocols and cross-study comparability. This hampers the quantification of S-R-E relationships and limits the development of predictive models and standardized design strategies. Structural parameters of MNBs vary widely, including size distribution, shell materials (lipid, polymer, protein) [12,13,14], gas core composition [12], and ligand density [18]. Even small differences in these factors can significantly impact acoustic response. Traditional preparation methods often suffer from batch-to-batch variability, reducing reproducibility, while newer techniques such as microfluidics improve consistency [25] but are not yet widely adopted. Acoustic parameters—including frequency, pressure, and duty cycle—also differ greatly, making direct comparisons difficult. For example, Kim et al. reported optimal drug release at 1 MHz/0.5 MPa [43], whereas Lentacker et al. found better results at 3 MHz/0.7 MPa [44]. Biological models also vary, with differences in cell lines, animal models, and cancer types (e.g., breast [40] vs. pancreatic [41]), each with distinct TMEs and barriers. Outcome measures are equally inconsistent, ranging from intracellular drug concentration [33] to reversal of MDR [6], while clinical studies often focus on survival or response rates [41]. These inconsistencies hinder data integration and limit the comparability and generalization of findings.

#### 5.1.3. Therapeutic Efficacy Challenge: Low In Vivo Delivery Efficiency and Unknown Long-Term Safety

In terms of efficacy evaluation, the first challenge is the uncertainty of in vivo pharmacokinetic behavior. Most microbubbles are cleared by the reticuloendothelial system within 30 min of injection, limiting their effective accumulation at the tumor site [45]. Additionally, the drug’s physicochemical properties significantly influence its loading and release efficiency [13]. Critically, data on long-term immunocompatibility and safety are limited, and standardized evaluation methods are lacking [39], which constitutes a major obstacle to clinical translation.

### 5.2. Future Research Directions

Future studies should first address data heterogeneity by standardizing experimental procedures and reporting methods. This includes consistent characterization of MNBs (e.g., size distribution and shell composition) and standardized reporting of acoustic parameters and outcomes. A shared framework for selecting biological models is also essential. With comparable baseline data, the S-R-E framework can move beyond qualitative analysis and become a reliable, predictive tool. This requires building physics-based mathematical models to predict the acoustic response of specific microbubble structures, thereby enabling the rational design of carriers and providing a theoretical basis for the standardization of acoustic parameters [30]. Another key direction is developing intelligent control systems that integrate AI (artificial intelligence)-driven real-time feedback. Such systems could use ultrasound imaging to monitor cavitation activity and enabling predictive modeling and adaptive therapeutic strategies. This would ultimately achieve personalized precision therapy that maximizes efficacy while minimizing side effects [24].

## 6. Conclusions

Cancer therapy is moving toward more precise and personalized approaches. Ultrasound combined with MNBs offers controllable activation, sonoporation, and real-time imaging, which together enhance drug targeting and tissue permeability.

From an engineering optimization perspective, this review introduces a “S-R-E” framework to systematically organize the intrinsic connections among bubble structural design, acoustic response modulation, and final therapeutic efficacy. Our review identified a key barrier to progress in this field—high data variability and the absence of standardized parameters. This issue directly limits the comparability of results across studies and restricts the potential of the S-R-E framework for quantitative modeling. We also summarize the role of advanced technologies like microfluidics and cell membrane biomimicry in achieving controllable fabrication and improved biocompatibility. Despite progress in preclinical and early clinical studies, translation remains limited by trade-offs between structural stability and sensitivity, lack of parameter standardization, and insufficient long-term safety data.

Future work should prioritize standardization, predictive modeling, and AI-based acoustic control with real-time feedback. These steps are essential to make micro/nanobubble technologies more standardized, predictable, and clinically applicable.

Overall, the ultrasound-responsive MNBs technology is not merely a drug delivery method but a multifunctional therapeutic platform integrating principles from physics, biology, and engineering. It shifts the paradigm from passive drug accumulation to active, controllable delivery, offering a new pathway for cancer treatment. Continued advances in mechanism studies, material optimization, and safety evaluation may enable this system to move from laboratory research to standardized clinical use, driving the progress of precision medicine.

## Figures and Tables

**Figure 1 pharmaceutics-17-01378-f001:**
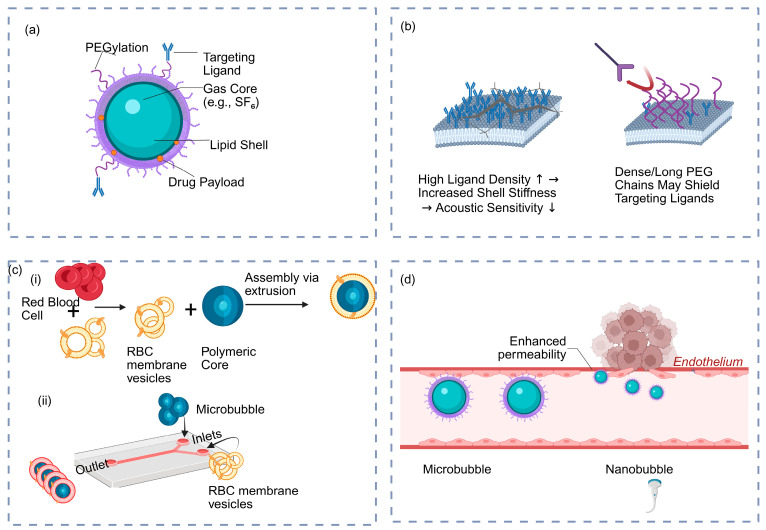
Schematic overview of the structural engineering of MNBs. (**a**) Fundamental components of a drug-loaded and surface-functionalized microbubble. (**b**) Critical design trade-offs between targeting efficacy and acoustic sensitivity. The left panel shows that a high density of targeting ligands (↑) leads to (→) increased shell stiffness, which in turn results in (→) decreased acoustic sensitivity (↓). The right panel illustrates how dense PEG chains can shield targeting ligands (blue Y-shapes) from binding to their corresponding target receptor on a tumor cell (purple Y-shape). (**c**) Advanced fabrication strategies, including (**i**) biomimetic membrane coating and (**ii**) microfluidic fabrication, where the arrows indicate the injection of precursor materials (microbubbles and Red Blood Cell membrane vesicles) into the device inlets for controlled assembly. (**d**) Size-dependent biodistribution, comparing the intravascular confinement of MBs to the extravasation of NBs.

**Figure 2 pharmaceutics-17-01378-f002:**
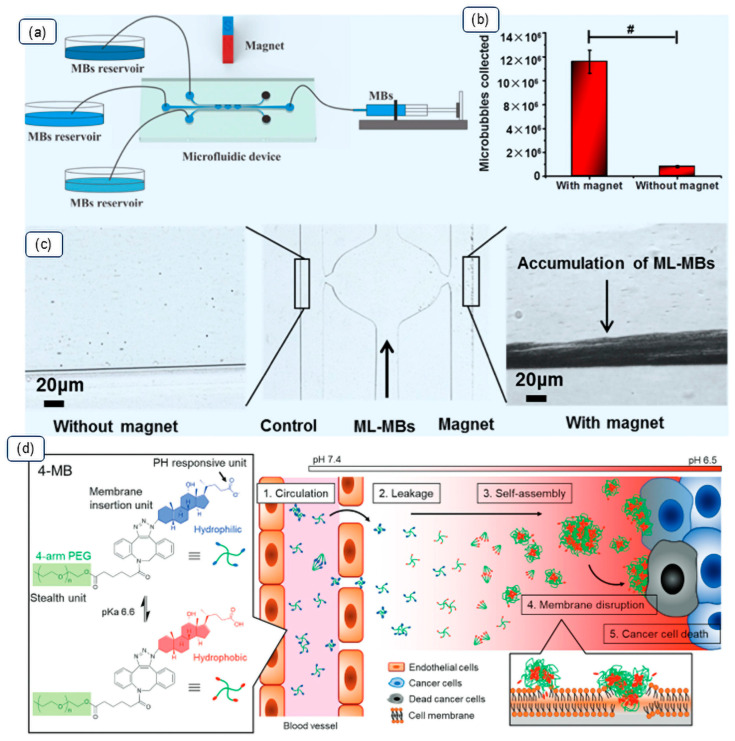
Advanced microbubble design strategies. (**a**–**c**) Magnetic targeting of doxorubicin-loaded magnetic lipid microbubbles in a microfluidic device. (**a**) Schematic of the microfluidic chip. (**b**) Quantification confirming increased ML-MBs retention with a magnetic field; the ‘#’ symbol indicates a statistically significant difference (e.g., *p* < 0.05). (**c**) Microscopic image showing magnetic-guided accumulation in the channel, where the vertical arrow indicates the direction of fluid flow. (**d**) A pH-responsive microbubble (4-MB) triggered by the acidic tumor microenvironment (pH 6.5). In the 4-MB structural diagram, the colored components and corresponding text represent the hydrophilic pH-responsive unit (blue), the hydrophobic membrane insertion unit (red), and the 4-arm PEG stealth unit (green). The curved arrows illustrate the sequential therapeutic process: (1) circulation, (2) leakage, (3) self-assembly into nanoparticles, (4) membrane disruption, and (5) subsequent cancer cell death. Panels (**a**–**c**) adapted with permission from Ref. [15]. Copyright 2020, American Chemical Society. Panel (**d**) adapted with permission from Ref. [21]. Copyright 2021, The Authors.

**Table 1 pharmaceutics-17-01378-t001:** A Comparison of the Physicochemical and Biological Properties of Microbubbles and Nanobubbles. Data compiled from references [25,26,27,28].

Parameter	Microbubbles	Nanobubbles
Size	1–10 µm (typically >1 µm)	<1000 nm (typically <500 nm)
In Vivo Distribution	Strictly intravascular. Too large to penetrate the tumor vessel wall and enter the tumor stroma	Can extravasate. Their nanoscale size allows them to pass through the 380–780 nm gaps in the tumor vessel wall to achieve extravascular drug delivery
Acoustic Response	Strong acoustic response. For an equivalent gas volume, their larger size generally results in a superior ultrasound signal enhancement compared to NBs	Weaker acoustic response. The smaller size reduces the acoustic response, making high-quality ultrasound imaging difficult to achieve
Circulation & Retention	Short circulation time, poor tumor retention. In animal models, their signal intensity in the tumor region attenuates rapidly after injection	Long circulation time, good tumor retention. Benefiting from the EPR effect, their signal intensity in the tumor area persists significantly longer than that of MBs, demonstrating superior passive targeting
Key Clinical Applications	1. Blood pool contrast agents 2. Transient opening of physiological barriers (e.g., BBB) 3. Enhancing drug permeation through the vascular endothelium	1. Delivering drugs deep into the extravascular tumor stroma 2. Tumor-targeted imaging and therapy via the EPR effect 3. Gene delivery to deep tissues

Abbreviations: BBB, blood–brain barrier; EPR, enhanced permeability and retention; MBs, microbubbles; NBs, nanobubbles.

## Data Availability

No new data were created or analyzed in this study.

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
