# Peer review of "A Review on the Structure-Response-Efficacy Optimization of Ultrasound-Responsive Micro/Nanobubbles for Cancer Therapy"

_pharmaceutics, 2025, doi:10.3390/pharmaceutics17111378_

Round 1

Reviewer 1 Report

Comments and Suggestions for Authors

Comments to the Authors:

This short and concise review summarizes crucial aspects of using nanobubbles and microbubbles in cancer therapy with ultrasound. The text is generally clear, and the limited length is not a problem in itself.

1) I have a major concern: there are numerous existing reviews on bubbles and ultrasound, including very new meta-reviews and historical perspectives (see e.g., https://doi.org/10.1016/j.addr.2024.115199

 and https://doi.org/10.1016/j.addr.2023.115178

). These are easy to find in scientific search engines. This manuscript largely repeats content available elsewhere, so I am not convinced of its novelty.

As a general comment, I recommend focusing on more specific optimization strategies (the general statements included are fine, but insufficient). It may be better to elaborate on the already raised issue, e.g., the challenge of optimization when experimental conditions differ widely and bubble-based platforms can be complex (because different authors often test novel designs). Without this, the review is acceptable but unlikely to engage readers of Pharmaceutics. The manuscript needs to be reviewed, extended, and corrected.

More detailed comments are as follows.

2) In the abstract, the specific contribution of this review should be emphasized more clearly to distinguish it from the sources it summarizes.

3) The graphical abstract is of low quality. Without additional clues, it is not clear what panels a–e depict. Please improve it for readers.

4) In general, the quality of all figures should be improved significantly.

5) In line 44, the authors repeat what is stated in the abstract, that MNBs are non-invasive. This is a broad statement that needs clarification. What exactly is meant by “non-invasive,” especially given that issues with intravenous administration are discussed extensively in the literature, although not in the current review (it might be another point to extend).

6) Abbreviations should be defined only upon first use (e.g., the US description in line 62 is unnecessary, as “ultrasound” appears earlier). There are several similar examples; please correct them throughout the manuscript.

7) Panels a–d in Figure 1 are totally unclear in the PDF file. Although the caption explains them, the panel text is too small to be useful. Consider splitting Figure 1 into two figures or remaining only panels d and e (the others can be described briefly in the text).

8) The end of the Figure 1 caption (line 84) is not understandable. Please check it.

9) There appears to be a technical problem with subsection titles (e.g., the title is missing for Section 2.1, line 85). Please check it carefully.

10) In ultrasound research, experimental conditions are crucial. For example, line 100 mentions 0.6 MPa, but it is not specified whether this is an rms value, a peak negative/positive pressure, etc. Please provide detailed acoustic parameters from the cited papers (line 100 is only one example, as the same problems appear elsewhere).

11) The sentence in lines 101–102 is unclear and should be rewritten.

12) Theranostics is a fast-growing area in drug delivery research. It is mentioned briefly and in a somewhat narrow context, specifically in relation to reduced toxicity (line 119, Section 2.3). There are many reports, including reviews, on theranostic aspects of bubbles; I suggest adding a dedicated subsection and describing the ideas in the reviews (as e.g., see https://doi.org/10.7150/thno.62218

, https://doi.org/10.7150/thno.70372

, https://doi.org/10.1002/anie.202422278).

13) I am not sure “in summary” is appropriate at the end of the subsection (line 234). Consider rephrasing it.

14) Generally, the layout of the whole is clear. The authors note that AI was used for grammar improvement, so the text reads smoothly. However, some subsection content remains unclear. For instance, in Subsection 4.2, the Authors discuss surface functionalization to improve targeting specificity, which was already covered earlier when describing ref. 32.

15) In line 264, a theranostic aspect of another work is recalled (the need for a dedicated theranostics subsection).

16) The composition of panels in Figure 4 is also unclear, especially panel labels are too small. As with Figure 1, I recommend limiting the number of panels to the most important.

17) The review addresses ultrasound bubbles and some physical aspects, but the context is sometimes strange. For example, line 313 refers to studies showing that acoustic parameters influence acoustic cavitation, which is unsurprising and very general. In the same paragraph, the pressure value is again undefined. A similar overly general statement appears in lines 323–324.

18) Figure 5 should be rearranged to show axis labels in each panel clearly. The number of panels is excessive again, in my opinion.

19) Chapter 5 is important. However, Subsection 5.4 is too general and too brief. It is widely known that comparing experimental data across studies is challenging in the natural sciences. Citing only two examples from randomly chosen papers is insufficient. It would really benefitial for the readers if the Authors conducted a broader survey to show a comparative set of results illustrating the diversity of conditions, as mentioned. In the same paragraph (line 394), AI is mentioned without linking to the literature or making clear that this is the Authors’ opinion.

20) The paragraph at line 422 seems not necessary. Instead, more developed “future directions,” as introduced earlier, would be needed. This short review summarizes basic facts on using nanobubbles and microbubbles in cancer therapy with ultrasound. The text is understandable and the fact that it is a short review is not an issue.

Reviewer 2 Report

Comments and Suggestions for Authors

In this study, the authors mentioned the emerging role of ultrasound-responsive micro/nanobubbles (MNBs) in cancer therapy, focosing on the combination of ultrasound and MNBs, enhancing targeted drug and gene delivery through cavitation, sonoporation, and ultrasound-triggered site-specific release. Critical parameters including bubble size and shell composition, surface ligand modification, stability, pharmacokinetics, and acoustic optimization have been mentioned to improve therapeutic efficacy, considering targeting precision and immune evasion. Authors have explained the applications in chemotherapy, gene therapy, and image-guided treatment. The have included that MNBs increase local drug concentration and reduce systemic toxicity. Bubble stability and acoustic sensitivity, variability are challenges that need to be standardized in protocols and clinical validation. There are promising future directions for ultrasound-responsive micro/nanobubble (MNB) technology but challenges should be addressed for effective clinical translation. Overall, the manuscript would be strengthened by integrating a more critical and analytical perspective on existing research challenges, clearer and more polished language, and a sharper focus on novel conceptual contributions or specific future research strategies. This would enhance its impact in the field of ultrasound-assisted drug delivery for cancer therapy.

However, there are some of issues to be addressed.

  1. In line 39, drug resistance due to MDR can be considered as another obstacle for drug efficacy.
  2. Increase the resolution of text in Graphical abstract.
  3. Authors are suggested to focus on "theranostic" word for MNMs applications.
  4. For 4.1. Bubble Structure and Shell Material Selection (line 199), authors can categorized the micro bubbles in terms of shell composition: protein, lipid and polymers and compare them.
  5. More ligands should be mentioned under 4.2. Surface Functionalization and Targeting Ligand Design.
  6. If there is, mention the biomimetic MNBs.
  7. Authors are suggested to mention more techniques for MNBs fabrication like microfluidic system. They should highlight the correlation between using MNBs as contrast agents/carrier and monodispersity.
  8. Please highlight the novelty of the study. The manuscript does not present novel experimental findings, focusing instead on a descriptive review. Many cited works are recent, but the paper largely integrates existing knowledge rather than offering new hypotheses or engineering concepts.
  9. Please other limitation like scalability.
  10. Unfortunately, authors ignore the role of gas types in microbubble core. Please mention them like SF6, Fluorocarbone, N2, …

Additional comments:

  • Authors should present new experimental data.
  • The discussion on in vivo distribution and pharmacokinetics highlights considerable variability but does not propose concrete new experimental approaches or modeling to address these gaps.
  • More discussion is needed about potential immunogenicity or long-term safety profiles of micro/nanobubbles, which is important for clinical translation.
  • Ultrasound parameters are noted as heterogeneous across studies, but the paper stops short of proposing standardized protocols or experimental frameworks to unify future research efforts.
  • While the manuscript is generally well-written, some sentences are overly complex or lengthy, reducing readability (e.g., long compound sentences in sections discussing engineering optimization could be split for clarity).
  • There are minor inconsistencies in abbreviation usage and occasional redundancies (e.g., microbubbles and nanobubbles sometimes abbreviated as MBsNBs without clear separation).
  • The phrasing in some parts could be more concise and direct, especially in the conclusion where points are repeated.
  • A few instances of awkward phrasing appear, e.g., "higher ligand density does not always correlate with improved targeting ligand density must be optimized" where a comma or conjunction is missing, affecting flow.
  • The future directions section outlines broad goals (personalized treatment, algorithmic control, clinical trials) that, while relevant, remain general and somewhat predictable without novel strategic insights or defined research roadmaps.
  • Suggestions for AI-based ultrasound parameter optimization or personalized therapy are noted but not elaborated in a way that advances current understanding or methodology uniquely.

Comments on the Quality of English Language

Moderate English editing is needed.

  • While the manuscript is generally well-written, some sentences are overly complex or lengthy, reducing readability (e.g., long compound sentences in sections discussing engineering optimization could be split for clarity).
  • There are minor inconsistencies in abbreviation usage and occasional redundancies (e.g., microbubbles and nanobubbles sometimes abbreviated as MBsNBs without clear separation).
  • The phrasing in some parts could be more concise and direct, especially in the conclusion where points are repeated.
  • A few instances of awkward phrasing appear, e.g., "higher ligand density does not always correlate with improved targeting ligand density must be optimized" where a comma or conjunction is missing, affecting flow.

Round 2

Reviewer 1 Report

Comments and Suggestions for Authors

The Authors addressed my comments on the layout of the manuscript and figures. They have divided the manuscript into subsections, making it more straightforward for readers. 

I am still not convinced of the novelty of this review, as the Authors did not address problematic issues (such as differences in experimental data sets across reports). I would suggest taking this problem into future research. 

Reviewer 2 Report

Comments and Suggestions for Authors

.

Comments on the Quality of English Language

Moderate English editing is needed.

  • While the manuscript is generally well-written, some sentences are overly complex or lengthy, reducing readability (e.g., long compound sentences in sections discussing engineering optimization could be split for clarity).
  • There are minor inconsistencies in abbreviation usage and occasional redundancies (e.g., microbubbles and nanobubbles sometimes abbreviated as MBsNBs without clear separation).
  • The phrasing in some parts could be more concise and direct, especially in the conclusion where points are repeated.
  • A few instances of awkward phrasing appear, e.g., "higher ligand density does not always correlate with improved targeting ligand density must be optimized" where a comma or conjunction is missing, affecting flow.
